# Variation in Body Composition Components Across Different Age Groups and Proposal of Age-Specific Normative Tables: A Cross-Sectional Study

**DOI:** 10.3390/nu17091435

**Published:** 2025-04-24

**Authors:** Kleber E. G. Barbão, Audrei Pavanello, Fabiano M. Oliveira, Natalia Q. Santos, Pablo Valdés-Badilla, Luciana L. M. Marchiori, Emerson Franchini, Braulio H. M. Branco

**Affiliations:** 1Graduate Program of Health Promotion, Cesumar University, Maringá 87050-390, Brazil; kleberbarbao@clinisport.com.br (K.E.G.B.); audrei.pavanello@unicesumar.edu.br (A.P.); profabiano.edu@gmail.com (F.M.O.); natquevedo01@gmail.com (N.Q.S.); lucianamarchiori@sercomtel.com.br (L.L.M.M.); 2Clinisport Prime, Integrated Rehabilitation and Performance Center, Maringá 87060-350, Brazil; 3Department of Physical Activity Sciences, Faculty of Educational Sciences, Universidad Católica del Maule, Talca 3530000, Chile; valdesbadilla@gmail.com; 4Sports Coaching Career, School of Education, Universidad Viña del Mar, Viña del Mar 2580022, Chile; 5School of Physical Education and Sport at the University of São Paulo, São Paulo 05508-030, Brazil

**Keywords:** reference standards, health policy, health promotion, public health

## Abstract

**Background/Objectives:** Utilizing a significative sample, this study aimed to analyze body composition components in different age groups and to develop age-specific normative tables for individuals in southern Brazil. **Methods:** This observational, descriptive, and cross-sectional study evaluated 8556 individuals of both sexes (54% females) aged 18–49. The hypotheses of the present study are related to declining fat-free mass (FFM), lean mass (LM), and skeletal muscle mass (SMM) and increasing fat mass (FM) and body fat percentage (BFP) during the aging process. Data were collected through bioelectrical impedance analysis (BIA) and stratified by age (18–29, 30–39, and 40–49 years), sex, and body mass index (BMI) classifications (normal weight, overweight, grade I, and grade II obesity). Following the comparison, body composition components were presented in the 3rd, 10th, 25th, 50th, 75th, 90th, and 97th percentiles. **Results:** This study’s main findings indicated that FM and BFP increased with age in both sexes. Among overweight and obese individuals, elevated BFP was particularly high in obese females aged 40–49 years and in normal and overweight males. FFM, LM, and SMM were generally lower in the 40–49-year-old group, although obese females over 40 presented higher FFM and LM values. In contrast, males presented lower FFM and LM values but higher values among individuals with higher BMI. SMM was lower in overweight individuals over 40, likely reflecting muscle mass loss associated with aging. **Conclusions:** Based on these results, lifestyle interventions that combine nutrition and physical exercise may be recommended to mitigate these effects of aging.

## 1. Introduction

Body composition analysis provides a detailed assessment of body components, distinguishing between body fat percentage (BFP) and fat-free mass (FFM), including skeletal muscle mass (SMM), which is crucial for understanding cardiometabolic and cancer risks [1]. In this sense, BFP is a vital metric for assessing body composition, enabling the identification of health risks associated with excessive values of fat mass (FM) [2,3]. Recent studies underscore the relationship between high BFP and chronic diseases (non-communicable diseases—NCDs), such as type 2 diabetes mellitus (T2DM) and cardiovascular diseases (CVDs) [2,3].

Furthermore, monitoring BFP is relevant for the efficacy of physical exercise training and nutritional programs, assisting in body mass control and muscle hypertrophy [4]. Precise assessment methods, such as bioelectrical impedance analysis (BIA), effectively determine BFP and FFM and can be valuable for health professionals and trainers [5,6]. BFP is an essential health indicator [7,8], and appropriate fat levels can prevent the development of NCDs [9].

Additionally, BFP directly influences health-related physical fitness, affecting muscle strength, endurance, and physical performance, with implications for overall health status and quality of life [10]. Modern society faces significant challenges due to low physical activity levels and high consumption of processed foods, contributing to increasing obesity rates and NCDs [11], which may be linked to comorbidities [11].

Ponti et al. [11] reported that aging is characterized by significant changes in body composition, including an increase in FM, particularly visceral fat, and a decrease in lean mass (LM), known as sarcopenia (adding to loss of function). These alterations contribute to a heightened risk of chronic diseases, such as cardiovascular diseases and T2DM. The redistribution of body fat, accumulation in the trunk, and loss of SMM negatively impact the functional capacity of older people. Understanding these changes is crucial for developing personalized preventive strategies to improve health, fitness, performance, and quality of life, particularly in old age [11]. In addition, systematic meta-nalyses have identified that exercise and nutrition approaches could improve body composition components, i.e., increasing lean mass and reducing FM and BFP [12,13].

Considering the above mentioned aspects, establishing reference values for body composition components is essential for assessing health, fitness, and quality of life. Scientific studies suggest elevated body fat values are significantly associated with health risks, including cardiovascular and metabolic diseases [5,14,15]. Conversely, very low BFP can also be detrimental, negatively affecting reproductive function and immunity [14,15,16]. Therefore, identifying well-defined reference values enables the determination of ideal BFP ranges for different age groups and sexes, facilitating the guidance of healthy practices, fitness, and personalized interventions [17,18]. Moreover, establishing reference values for FFM, LM, and SMM can also aid health professionals in directing multidisciplinary approaches to improve health and quality of life for individuals with muscle weakness, muscle mass loss complaints, or even those seeking to enhance aesthetic, health, or performance aspects [18,19,20,21].

Thus, the primary aim of this study was to analyze potential variations in body composition among young and middle-aged adults. Additionally, acknowledging the importance of understanding and evaluating the specific needs of different population groups [5,14,15], the secondary aim was to create a normative table for classifying body composition components in Brazilians across different sexes and ages. These tables may help provide valuable awareness, implying the possibility of analyzing the responses to health promotion actions, such as physical exercise and nutritional interventions. The hypotheses of the present study are related to declining FFM, LM, and SMM and increasing FM and BFP during the aging process, factors that justify the implementation of normative tables in different age groups.

## 2. Materials and Methods

### 2.1. Study Design

This study employed an observational, descriptive, cross-sectional, comparative, and quantitative design [22], following the Strengthening the Reporting of Observational Studies in Epidemiology (STROBE) Statement [23]. It employed a cluster sampling method utilizing three clusters: the Municipal Hospital, the “Acclimation” Basic Health Unit, and the Interdisciplinary Laboratory for Intervention in Health Promotion. The response rate was similar across the groups, as we recruited the 11,554 participants homogeneously, with 1 participant each from the Municipal Hospital, the Basic Health Unit, and the Interdisciplinary Laboratory, continuing this pattern until 8556 eligible participants were identified. Consequently, 3108 participants were deemed ineligible based on exclusion criteria (as shown in Figure 1).

### 2.2. Statistical Power Analysis

A power analysis with the following parameters was performed to ensure adequate statistical power: 80% power was used to detect the minimum effect size (f = 0.10) in the analysis of variance (ANOVA), including 18 groups at an alpha level of 0.05. The ‘18 groups’ refer to the different combinations of age and BMI categories across sexes. Specifically, the groups were as follows: individuals aged 19–29 with normal BMI, overweight, and obesity; individuals aged 30–39 with the same BMI categories; and individuals aged 40–49 with the same conditions. Each age and BMI category was further divided by sex (male and female), resulting in the 18 distinct groups analyzed. The resulting minimum required sample size was calculated to be 1985 individuals. The analysis was performed in R (version 4.4.2) and RStudio (version 2024.09.1+394).

### 2.3. Participants

A total of 11,654 individuals were recruited. During the initial screening, participants were selected based on being within the appropriate age range and expressing interest in the study. At this stage, we did not conduct interviews related to personal histories or pre-existing conditions. Following this initial screening, 1596 individuals were excluded, leaving 10,058 participants. In the second screening, a more detailed evaluation was conducted, which led to excluding individuals over 50. This decision was made because the analysis lacked sufficient power to include these participants, resulting in 1452 additional exclusions. The remaining 8556 individuals were of both sexes and aged 18–49. The sample included 4628 females divided into age groups: 18–29 years (n = 2179), 30–39 years (n = 1237), and 40–49 years (n = 1212). The male sample consisted of 3928 participants, who were also divided into age groups: 18–29 years (n = 2069), 30–39 years (n = 1098), and 40–49 years (n = 761).

For this study, inclusion and exclusion criteria were established to ensure sample homogeneity and reliability. Eligible participants were adults of both sexes aged 18–49 years. Only individuals with stable health and without uncontrolled non-communicable diseases (NCDs), such as type 2 diabetes mellitus (T2DM) or hypertension, were included following a detailed health assessment. The sample comprised individuals with a body mass index (BMI) between 18.5 kg/m^2^ and 39.9 kg/m^2^, ranging from normal body mass to grade II obesity. Additionally, individuals with varying physical activity levels were included, from sedentary to those engaged in recreational and amateur sports. The short version of the International Physical Activity Questionnaire (IPAQ) assesses physical activity levels, ensuring precise participant classification [24].

The exclusion criteria were pregnant and menstruating females (during assessment), individuals with pacemakers or other implantable electronic devices, patients with paraplegia or quadriplegia, cancer, or post-bariatric surgery. Additionally, patients with uncontrolled chronic diseases were excluded, specifically those with T2DM or type 1 diabetes mellitus, defined by fasting glycemia ≥ 126 mg/dL. Other criteria included hypertension (blood pressure ≥ 130/80 mmHg), renal failure (GFR < 15 mL/min), or severe liver disease (diagnosis of cirrhosis), with all relevant exams conducted in advance. A simple interview was conducted to differentiate between type 1 and type 2 diabetes, focusing on age at diagnosis and treatment history. This ensured clearer categorization for exclusion. The final sample did not include individuals meeting any of these criteria. The inclusion and exclusion criteria are summarized in Figure 1.

### 2.4. Ethical Considerations

All study participants provided their consent by signing the informed consent form per the guidelines described in Resolution 466/2012 of the National Health Council of the Ministry of Health and the ethical principles described in the Declaration of Helsinki. The Local Ethics and Research Committee approved this study. All the participants signed the informed consent form and were informed about all the procedures performed in the research.

### 2.5. Data Collection

Data were collected over seven years, from 2017 to 2024, in Maringá, Paraná, in Brazil’s southern region.

### 2.6. Anthropometry Protocol

Following Lohman, Roche, and Martorell [25], the following procedures were adopted to obtain the anthropometric parameters. Each participant’s height was measured barefoot, with the shoulders, buttocks, and head in contact with the wall, with heels together and touching the base of the stadiometer. When the participant’s head was perpendicular to the stadiometer, a movable headboard was placed on it. The participants looked forward, keeping the chin level. After measurement, the height was recorded in meters (without rounding). The BMI was calculated based on the body mass values obtained via BIA and the height values.

### 2.7. Body Composition via Bioelectrical Impedance Analysis

To perform BIA, the participant must meet specific requirements: fast for 2–4 h before the examination, avoid vigorous physical activities the day before, urinate 30 min before the examination, abstain from alcohol 4 h prior, and avoid caffeine 12 h prior [5]. The InBody BIA (model 570, InBody Co., Seoul, Republic of Korea) was used, employing impedances at three different frequencies (5 kHz, 50 kHz, and 500 kHz) in each of the five body segments (right arm, left arm, trunk, right leg, and left leg) via 8-point tactile electrodes. Measurements were performed under standardized conditions, with ambient temperatures between 23 and 25 °C [26]. Total body water and body composition (FM, BFP, FFM, LM, and SMM) can be estimated through BIA [27].

### 2.8. Statistical Analysis

To determine differences in body composition parameters (FM, BFP, FFM, LM, and SMM) across sex and age groups and BMI categories, Welch’s one-way analysis of variance (ANOVA) was performed after normality confirmation via the Kolmogorov–Smirnov test. Post hoc pairwise comparisons were conducted via Holm’s adjustment, with the statistical significance set at *p* < 0.05. Age was categorized into three groups (18–29, 30–39, 40–49 years), and BMI was classified as normal, overweight, or obese. To address the impact of clustering on our sample, we calculated the Intraclass Correlation Coefficient (ICC) to reduce bias induced by the sampling. The effect was negligible for our outcome variables: FM, BFP, FFM, LM, and SMM (ICC = 0.000).

Linear regression analyses were performed for each body composition parameter to summarize the observed effects via the following formula: variable of interest~BMI + age group. BMI and age group were treated as categorical variables, with normal as the BMI reference level and 18 to 29 as the age group reference level. All analyses and visualizations were performed in R (version 4.4.2) and RStudio (version 2024.09.1+394) via the *ggstatsplot* package [28].

### 2.9. Establishing a Normative Table

Data were distributed in percentiles p3, p10, p25, p50, p75, p90, and p97 following the recommendations of Branco et al. [5]. The percentiles were selected to encompass the entire Gaussian curve, with p50 representing the median, while p25 and p75 correspond to the interquartile range, where most of the sample data reside. Furthermore, p10 and p90 enable the analysis of data distribution near the extremities, which would not be possible to examine using central measures alone. In turn, p3 and p97 are considered extreme values, allowing for a more refined analysis of health parameters and performance metrics. FM, BFP, FFM, LM, and SMM were distributed in the percentiles. Statistical analyses were conducted via IBM SPSS Statistics ver. 20.0 software (IBM Co., Armonk, NY, USA).

## 3. Results

### 3.1. Descriptive Statistics

Table 1 presents the general characteristics of the male participants (n = 3928 participants) across the age groups: 18–29 years, 30–39 years, and 40–49 years.

Table 2 presents the general characteristics of the females (n = 4628 participants), distributed by age group and across age ranges, i.e., 18–29 years, 30–39 years, and 40–49 years.

### 3.2. Body Composition Comparison According to Age and BMI

Greater fat mass (FM) was observed with increasing age in overweight and obese females (Figure 2A, *p* < 0.01) and in normal and overweight males (Figure 2B, *p* < 0.001), but not in obese males (Figure 2B, *p* < 0.01). The BFP presented a similar result, with higher BFP in older obese females (Figure 3A, *p* < 0.001). In males, there was a clear trend toward greater BFP in the normal and overweight groups (Figure 3B, *p* < 0.001), but not in obese males (Figure 3B, *p* < 0.001).

Figure 3 shows the BFP according to age, BMI, and sex of the population analyzed.

The FFM was greater in obese females (Figure 4A, *p* < 0.01) and lower in overweight males (Figure 4B, *p* < 0.001). LM had higher values in older obese females (Figure 5A, *p* < 0.01) and lower values in normal and overweight males (Figure 5B, *p* < 0.05). A similar pattern was observed for SMM, with lower values in older overweight females and males (Figure 6A,B, *p* < 0.01).

Figure 4 shows the FFM according to the age, BMI, and sex of the population analyzed.

Figure 5 shows the LM according to age, BMI, and sex of the population analyzed.

Figure 6 shows the SMM according to age, BMI, and sex of the population analyzed.

The coefficients for the body composition metrics in Figure 7A,B were calculated to summarize these findings. Females had higher FM according to age and BMI, with a similar pattern observed in BFP (Figure 7A). We also observed lower FFM and SMM in older female individuals and higher values of these parameters with higher BMIs (Figure 6A).

Higher values of FM were observed in males and older and overweight or obese individuals (Figure 7B). These findings were also reflected in BFP, where lower FFM, LM, and SMM values in older individuals were associated with higher values of these metrics among overweight and obese males (Figure 7B).

Table 3 presents the cutoff points for the body composition parameters of the Brazilian male adults included in the study.

Table 4 presents the cutoff points for the body composition parameters of the Brazilian female adults included in the study.

## 4. Discussion

This study analyzed body composition changes across various age groups, different BMI classifications (normal weight, overweight, and grade I and II obesity), and sexes. The main findings were as follows: (i) FM and BFP increased progressively with age in both sexes, with more pronounced effects in overweight and obese individuals; (ii) in females, higher values of FM and BFP were observed in older age groups, particularly among obese females aged 40–49 years; (iii) in males, there were noticeably higher values of FM and BFP in the normal and overweight groups, although not in obese males aged 40–49 years; (iv) FFM, LM, and SMM presented lower values in older groups, especially in those over 40 years; (v) females over 40 years with obesity presented higher values of FFM and LM, whereas males presented lower values of these parameters in the older groups but higher values with higher BMI; and (vi) SMM was lower in overweight individuals, likely reflecting the muscle mass loss associated with aging.

In addition, this study establishes reference values for various body composition parameters, including FM, BFP, FFM, LM, and SMM. This approach, which stratifies data by age and sex, allows for a detailed view of the specific values for each age group. Specific absolute values observed across ages and sexes provide relevant information for clinical practice and public health. The results of the present study revealed significant differences in body composition parameters among the 18–29-year-old, 30–39-year-old, and 40–49-year-old age groups, considering sex and BMI.

Overall, there were higher values of FM and BFP in older groups of both sexes, a result that was more evident in overweight and obese individuals. Females presented markedly higher values of BFP, particularly in older obese females, whereas males presented higher values of FM in the normal and overweight groups in the same age group. Conversely, progressively lower FFM, LM, and SMM values were observed in the older groups, except in older obese females, who presented higher values of these parameters at 40–49 years. These differences reflect the complex interplay between aging, BMI, and sex differences in body composition, underscoring the importance of personalized interventions to mitigate the adverse effects of aging on health [14].

Examining the absolute values for body composition variables between males and females revealed an apparent pattern of gradual differences in FM and BFP across age groups. The observed values suggest higher values for FM and BFP in older groups. This observed pattern aligns with the findings of Amaral et al. [29], who identified hormonal changes and reduced physical activity practices as factors potentially contributing to progressive fat accumulation. Additionally, Ofenheimer et al. [30] highlighted that the visceral adipose tissue in older age groups was associated with an elevated risk of developing metabolic diseases, such as T2DM and cardiovascular diseases, concurrent with higher values for BFP; the absolute values suggest a smaller FFM in older groups, particularly in the age groups above 40 years for both sexes. These observed differences in the data provide a descriptive overview of body composition changes across the lifespan, offering insights that may warrant further investigation. This approach to data analysis allows for a nuanced examination of body composition variables across different age groups and between sexes while maintaining a prudent interpretation of the observed patterns [14,28,29].

These findings align with Ofenheimer et al. [30], who documented the loss of LM associated with aging, a phenomenon exacerbated by sarcopenia. In line with this observation, Amaral et al. [26] emphasized that nutritional and physical activity interventions can mitigate this loss, although this is challenging. The decline in FFM is accompanied by reductions in LM and SMM, particularly in older individuals. As elucidated by Ofenheimer et al. [30], decreasing SMM is related to a decline in anabolic hormone production and decreased physical activity, reinforcing the importance of these findings. Amaral et al. [29] emphasized maintaining SMM to preserve functionality and independence, highlighting the need for preventive strategies (physical exercise, mainly strength training plus adequate nutritional support).

The lower FFM values in the older age groups likely reflect decreased LM and increased BFP due to aging. Amaral et al. [29] and Ofenheimer et al. [30] agreed that a decrease in FFM is an essential marker for the early identification of sarcopenia, especially in individuals between 50 and 59 years of age. This age group is particularly vulnerable to declines in physical functionality and increased risk of NCDs, making monitoring FFM as part of a comprehensive body composition assessment even more evident. Classification tables derived from studies such as this play a crucial role in health, especially for professionals dedicated to nutrition and physical conditioning [31]. As a final point, given changes in the aging process, Deutz et al. [32] and Thomas et al. [33] point out that regular exercise, particularly resistance training, can normalize some aspects of age-related mitochondrial dysfunction and improve muscle function. The combination of protein nutrition and exercise is optimal for maintaining muscle function, preventing frailty, sustaining independence during aging, and promoting health [32,33]. Finally, in systematic review studies, Xie et al. [12] and Egleseer et al. [13] discuss that exercise and nutrition interventions could improve body composition components, reduce FM and BFP, and improve LM, leading people to a higher quality of life.

## 5. Limitations and Strengths of the Study

Identifying indicators of body composition, such as differences in FFM, is crucial for implementing effective prevention and intervention strategies to maintain health and well-being. The sample was restricted to a specific region of Brazil, potentially limiting the generalizability of the findings to other populations. Additionally, this was a cross-sectional study in which causality between variables could not be analyzed. Future research should address these limitations by expanding the age range studied to include children, adolescents, and individuals over 50 years, which is essential for developing specific normative tables for these diverse populations.

Analyzing specific subgroups, such as athletes or individuals with health conditions, could further create more detailed normative tables tailored for these groups. Longitudinal studies are equally important in tracking changes in body composition over time and better understanding the factors influencing these changes. By following these recommendations, future research is expected to advance the knowledge of body composition and its application in promoting health and well-being.

In addition, this research has several strengths, as follows: a large sample size; the use of males, females, and different age ranges, which allows analysis of the results according to the characteristics of the population; the establishment of cutoff points for body composition through objective methods with high reliability, such as BIA; and the provision of relevant input for the clinical field, such as a reference table that allows the establishment of parameters for different populations.

## 6. Practical Applications

These tools are fundamental in allowing such professionals to direct nutritional and physical activity interventions in a personalized manner, ensuring that the individual needs of each patient or client are effectively met [31]. Using these reference tables, health professionals can accurately identify areas needing special attention, whether for weight loss or muscle hypertrophy, among other health and well-being objectives [31]. Regularly monitoring body composition indicators via classification tables as a reference is crucial in evaluating the effectiveness of nutritional and training interventions, allowing precise and timely adjustments to optimize results. Body composition can directly influence the performance of these groups’ post-exercise recovery and injury risk [33]. The impacts of this study on health promotion during aging are substantial. Interventions aimed at maintaining LM and reducing excess fat can improve mobility, reduce the risk of falls, and enhance independence in older age, and these reference tables could help health professionals access and drive assertive actions to improve body composition components in early-adult age groups.

## 7. Conclusions

The findings indicated progressively higher values of FM and BFP with advancing age in both sexes, a pattern that was more pronounced in overweight and obese individuals. In females, higher values of FM and BFP were observed with age, particularly among obese females aged 40–49 years. In males, there were higher values of FM and BFP in the normal and overweight groups than in the normal weight group, although this was not the case in obese males. Lower FFM, LM, and SMM values were observed in the older groups, especially those over 40. However, females over 40 years of obesity presented higher values of FFM and LM, whereas males presented lower values of these parameters with age but higher values among those with higher BMIs. Therefore, following the scientific literature, personalized interventions are recommended to mitigate the adverse effects of aging on health, and strength training plus adequate nutritional support may improve body composition status. Normative tables for analyzing body composition components can help health professionals monitor possible developments or even involutions during personalized interventions.

## Figures and Tables

**Figure 1 nutrients-17-01435-f001:**
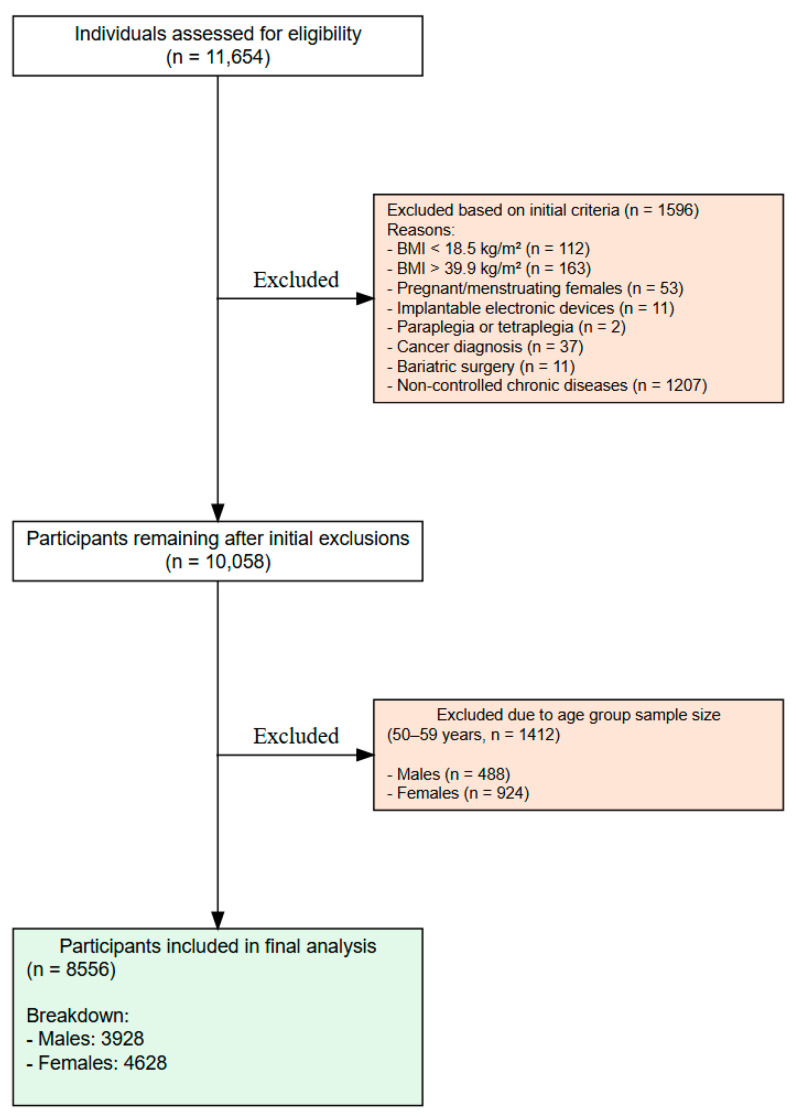
STROBE flowchart of participants of the study.

**Figure 2 nutrients-17-01435-f002:**
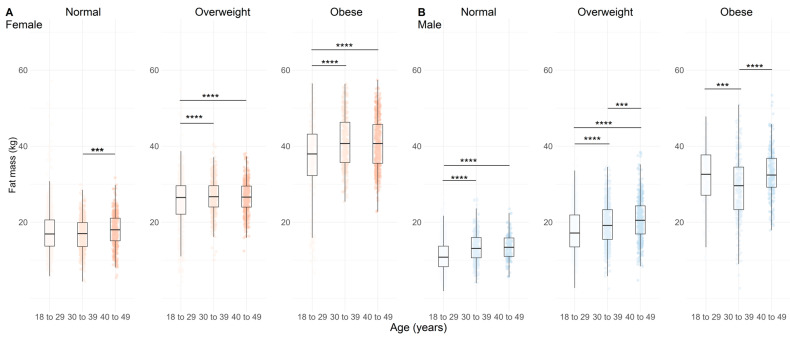
Fat mass distribution across age categories and BMI categories (normal, overweight, and obese) in females (**A**), n = 4628) and males (**B**), n = 3928). Data are represented by boxplots (median and interquartile range) with individual points overlaid for visualizing variability; n = 4628. Shades of red indicate age groups in females, while shades of blue indicate age groups in males. Statistical significance between age categories within each BMI group is indicated by adjusted *p-*values (Holm-adjusted), where *** = *p* < 0.01 and **** = *p* < 0.001.

**Figure 3 nutrients-17-01435-f003:**
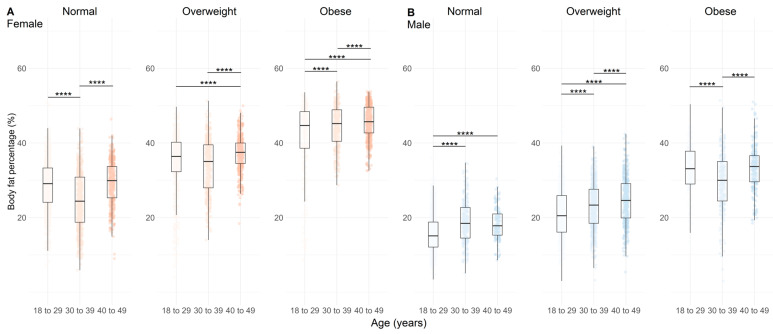
Body fat percentage distributions across age categories and BMI categories (normal, overweight, and obese) in females (**A**), n = 4628) and males (**B**), n = 3928). Data are represented by boxplots (median and interquartile range) with individual points overlaid for visualizing variability; n = 4628. Shades of red indicate age groups in females, while shades of blue indicate age groups in males. Statistical significance between age categories within each BMI group is indicated by adjusted *p* values (Holm-adjusted), where **** = *p* < 0.001.

**Figure 4 nutrients-17-01435-f004:**
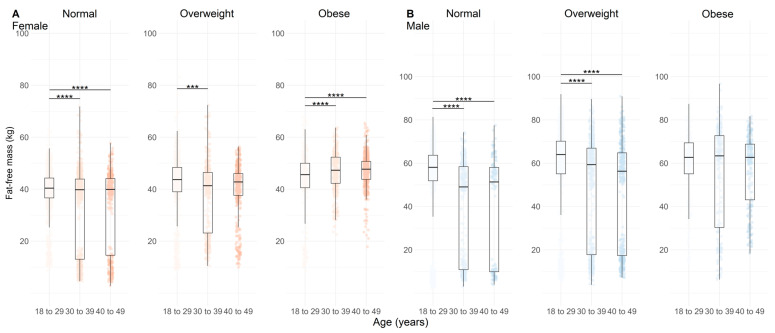
Fat-free mass distribution across age categories and BMI categories (normal, overweight, and obese) in females (**A**), n = 4628) and males (**B**), n = 3928). Data are represented by boxplots (median and interquartile range) with individual points overlaid for visualizing variability; n = 4628. Shades of red indicate age groups in females, while shades of blue indicate age groups in males. Statistical significance between age categories within each BMI group is indicated by adjusted *p* values (Holm-adjusted), where *** = *p* < 0.01 and **** = *p* < 0.001.

**Figure 5 nutrients-17-01435-f005:**
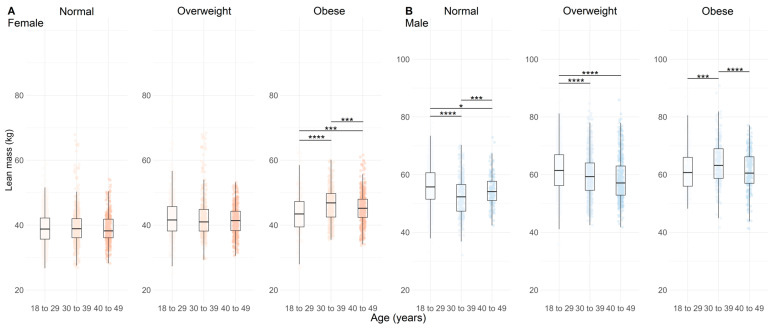
Lean mass distribution across age categories and BMI categories (normal, overweight, and obese) in females ((**A**), n = 4628) and males ((**B**), *n =* 3928). Data are represented by boxplots (median and interquartile range) with individual points overlaid for visualizing variability; n = 4628. Shades of red indicate age groups in females, while shades of blue indicate age groups in males. Statistical significance between age categories within each BMI group is indicated by adjusted *p* values (Holm-adjusted), where * = *p* < 0.05, *** = *p* < 0.01, and **** = *p* < 0.001.

**Figure 6 nutrients-17-01435-f006:**
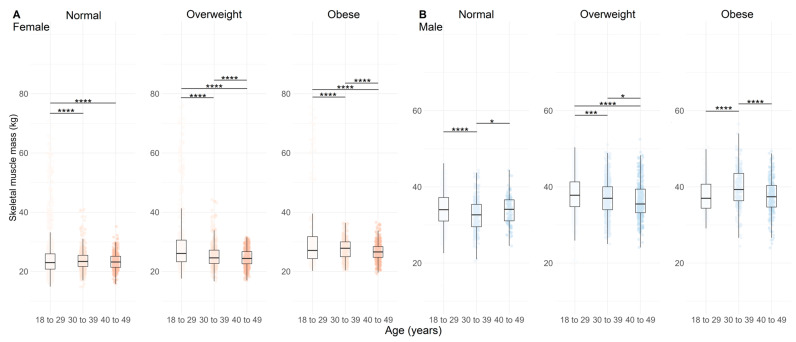
Distribution of skeletal muscle mass across age categories and BMI categories (normal, overweight, and obese) in females ((**A**), *n =* 4628) and males ((**B**), n = 3928). Data are represented by boxplots (median and interquartile range) with individual points overlaid for visualizing variability; n = 4628. Shades of red indicate age groups in females, while shades of blue indicate age groups in males. Statistical significance between age categories within each BMI group is indicated by adjusted *p* values (Holm-adjusted), where * = *p* < 0.05, *** = *p* < 0.01, and **** = *p* < 0.001.

**Figure 7 nutrients-17-01435-f007:**
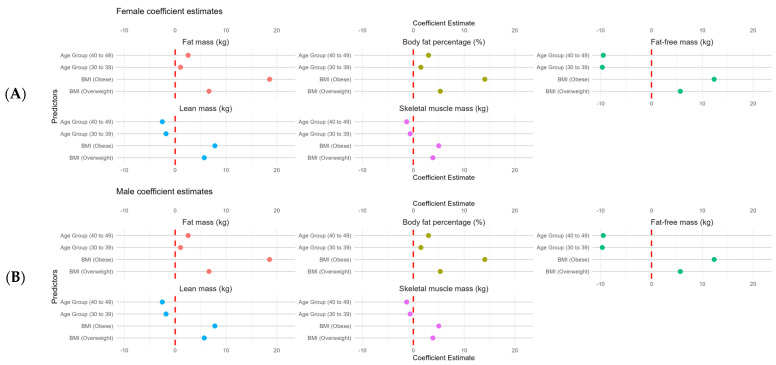
Coefficient estimates from linear regression models for body composition metrics (body fat mass, body fat percentage, fat-free mass, lean mass, and skeletal muscle mass) in females ((**A**), n *=* 4628) and males ((**B**), n = 3928), stratified by age group (30–39 years and 40–49 years, with 18–29 years as reference) and BMI category (overweight and obese, with normal as reference). The red dashed line indicates a coefficient of 0, indicating no effect. Positive coefficients indicate increases in the respective body composition metric relative to the reference category, whereas negative coefficients indicate a decrease. All the coefficients were statistically significant (*p* < 0.05).

**Table 1 nutrients-17-01435-t001:** The anthropometric characteristics of the anthropometric indices and body composition of the male adults included in the present study.

Males (*n =* 3928)	18–29 Years (*n =* 2069)		30–39 Years (*n =* 1098)		40–49 Years (*n =* 761)	
Variables	Mean ± SD	Min–Max	Mean ± SD	Min–Max	Mean ± SD	Min–Max
Body mass (kg)	79.6 ± 11.8	51.3–115.3	83.9 ± 12.5	51.6–110.0	87.6 ± 11.3	55.1–109.9
Height (cm)	176.7 ± 7.1	150.0–206.0	175.6 ± 6.6	152.0–199.0	175.5 ± 6.5	150.0–196.0
BMI (kg/m^2^)	25.5 ± 3.4	18.5–37.7	27.1 ± 3.5	18.5–39.3	28.2 ± 3.3	19.3–39.0
FM (kg)	16.2 ± 8.2	1.9–53.8	19.9 ± 8.2	2.5–50.9	23.0 ± 8.7	4.7–53.4
BFP (%)	19.7 ± 7.9	3.0–51.6	23.3 ± 7.4	3.0–51.4	26.0 ± 7.8	5.4–51.0
FFM (kg)	54.3 ± 21.7	2.1–95.5	46.3 ± 24.7	3.0–97.0	47.8 ± 23.4	3.9–90.9
LM (kg)	59.0 ± 7.9	26.3–89.9	58.6 ± 8.4	32.1–90.9	58.7 ± 7.4	41.3–85.9
SMM (kg)	36.1 ± 5.1	14.1–55.6	36.4 ± 5.3	20.5–56.5	36.3 ± 4.7	24.0–52.5

Note: Data are presented as the mean, standard deviation (SD), minimum, and maximum values; min = minimum; max = maximum; BMI = body mass index; FM = fat mass; BFP = body fat percentage; FFM = fat-free mass; LM = lean mass; and SMM = skeletal muscle mass.

**Table 2 nutrients-17-01435-t002:** The anthropometric characteristics of the anthropometric indices and body composition of the female adults included in the present study.

Females (*n =* 4628)	18–29 Years (*n =* 2179)		30–39 Years (*n =* 1237)		40–49 Years (*n =* 1212)	
Variables	Mean ± SD	Min–Max	Mean ± SD	Min–Max	Mean ± SD	Min–Max
Body mass (kg)	65.7 ± 12.1	45.0–110.2	72.2 ± 14.5	45.9–110.0	74.6 ± 13.9	46.5–110.0
Height (cm)	163.9 ± 6.6	145.0–190.0	164.1 ± 6.2	147.5–195.0	162.2 ± 6.6	138.0–187.0
BMI (kg/m^2^)	24.6 ± 4.0	18.5–39.5	26.7 ± 5.0	18.6–39.9	28.3 ± 5.1	18.7–39.9
FM (kg)	21.7 ± 9.1	3.4–57.1	26.2 ± 11.0	4.4–56.4	29.2 ± 10.9	5.4–11.8
BFP (%)	31.8 ± 8.5	4.0–54.0	32.5 ± 10.6	5.9–56.5	37.9 ± 8.3	9.0–53.8
FFM (kg)	40.0 ± 10.8	5.6–83.0	38.1 ± 14.7	4.6–72.4	40.5 ± 12.3	2.8–65.4
LM (kg)	40.5 ± 5.7	25.2–78.1	42.2 ± 6.4	26.9–68.4	42.1 ± 5.2	28.2–61.6
SMM (kg)	27.6 ± 10.7	14.9–76.2	25.3 ± 4.0	14.8–44.3	24.9 ± 3.2	15.8–36.6

Note: Data are presented as the mean, standard deviation (SD), minimum, and maximum values; min = minimum; max = maximum; BMI = body mass index; FM = fat mass; BFP = body fat percentage; FFM = fat-free mass; LM = lean mass; and SMM = skeletal muscle mass.

**Table 3 nutrients-17-01435-t003:** Cutoff points for body composition parameters of Brazilian male adults included in the present study.

Percentiles	p3	p10	p25	p50	p75	p90	p97
Fat mass (kg) percentiles							
18–29 years	5.8–7.6	7.7–10.2	10.3–14.0	14.1–19.7	19.8–27.9	28.0–37.0	≥37.1
30–39 years	7.9–10.6	10.7–13.8	13.9–18.5	18.6–24.7	24.8–31.0	31.1–35.5	≥35.4
40–49 years	9.7–12.3	12.4–16.1	16.2–21.8	21.9–29.2	29.3–35.2	35.3–40.2	≥40.3
Body fat percentage percentiles							
18–29 years	8.6–10.8	10.9–13.7	13.8–18.0	18.1–24.1	24.2–30.7	30.8–38.3	≥38.4
30–39 years	10.5–13.8	13.9–17.7	17.8–22.8	22.9–28.1	28.2–33.1	33.2–35.9	≥36.0
40–49 years	13.1–16.0	16.1–19.8	19.9–25.7	25.8–31.6	31.7–36.2	36.3–41.1	≥41.2
Fat-free mass (kg) percentiles							
18–29 years	5.7–10.9	11.0–52.5	52.6–60.7	60.8–67.1	67.2–73.2	73.3–80.0	≥80.1
30–39 years	7.2–10.1	10.2–17.9	18.0–55.6	55.7–65.7	65.8–73.4	73.5–77.3	≥77.4
40–49 years	7.8–11.2	11.3–21.5	21.6–56.9	57.0–65.2	65.3–73.2	73.3–79.0	≥79.1
Lean mass (kg) percentiles							
18–29 years	45.1–49.4	49.5–53.6	53.7–58.5	58.6–63.7	63.8–69.6	69.7–75.5	≥75.6
30–39 years	44.7–47.2	47.3–52.3	52.4–58.4	58.5–63.9	64.0–69.7	69.8–72.9	≥73.0
40–49 years	46.7–49.6	49.7–53.0	53.1–57.5	57.6–63.6	63.7–69.6	69.7–74.4	≥74.5
Skeletal muscle mass (kg) percentiles							
18–29 years	27.4–29.9	30.0–32.6	32.7–35.9	36.0–39.2	39.3–42.7	42.8–46.7	≥46.8
30–39 years	26.8–29.3	29.4–32.7	32.8–36.3	36.4–39.9	40.0–43.5	43.6–45.3	≥45.4
40–49 years	28.1–30.2	30.3–33.1	33.2–35.8	35.9–39.0	39.1–43.3	43.6–45.3	≥45.4

Note: Data are presented in percentiles: p3 = percentile three; p10 = percentile ten; p25 = percentile twenty-five; p50 = percentile fifty; p75 = percentile seventy-five; p90 = percentile ninety; and p97 = percentile ninety-seven.

**Table 4 nutrients-17-01435-t004:** Cutoff points for body composition parameters of Brazilian female adults included in the present study.

Percentiles	p3	p10	p25	p50	p75	p90	p97
Fat mass (kg) percentiles							
18–29 years	9.3–11.9	12.0–14.9	15.0–19.9	20.0–26.8	29.9–34.1	34.2–43.8	≥43.9
30–39 years	10.6–13.1	13.2–17.6	17.7–24.3	24.4–33.4	33.5–43.1	43.2–50.3	≥50.4
40–49 years	11.8–16.2	16.3–20.8	20.9–27.9	28.0–36.9	37.0–45.1	45.2–50.8	≥50.9
Body fat percentage percentiles							
18–29 years	15.3–20.9	21.0–25.9	26.0–32.1	32.2–37.5	37.6–42.5	42.6–48.4	≥48.5
30–39 years	13.2–17.8	17.9–24.2	24.3–32.9	33.0–40.7	40.8–47.6	47.7–51.0	≥51.1
40–49 years	20.8–26.8	26.9–32.3	32.4–38.1	38.2–44.4	44.5–49.1	49.2–51.5	≥51.6
Fat-free mass (kg) percentiles							
18–29 years	12.2–20.3	20.4–25.9	26.0–32.1	32.2–37.5	37.6–42.5	42.6–48.4	≥48.5
30–39 years	7.4–12.3	12.4–31.6	31.7–41.8	41.9–47.3	47.4–53.2	53.3–60.2	≥60.3
40–49 years	7.9–15.7	15.8–38.6	38.7–43.7	43.8–47.9	48.0–51.6	51.7–55.0	≥55.1
Lean mass (kg) percentiles							
18–29 years	30.9–33.8	33.9–36.6	36.7–39.7	39.8–43.9	44.0–47.8	47.9–52.5	≥52.6
30–39 years	32.3–35.3	35.4–37.6	37.7–40.9	41.0–45.6	45.7–50.5	50.6–57.5	≥57.6
40–49 years	32.9–35.6	35.7–38.1	38.2–41.9	42.0–45.5	45.6–48.6	48.7–51.8	≥51.9
Skeletal muscle mass (kg) percentiles							
18–29 years	18.4–19.8	19.9–21.7	21.8–24.2	24.3–27.8	27.9–40.7	40.8–60.7	≥60.8
30–39 years	19.3–20.8	20.9–22.4	22.5–24.6	24.7–27.4	27.5–30.3	30.4–34.4	≥34.5
40–49 years	19.3–20.8	20.9–22.5	22.6–24.7	24.8–27.0	27.1–29.3	29.4–30.9	≥31.0

Note: Data are presented in percentiles: p3 = percentile three; p10 = percentile ten; p25 = percentile twenty-five; p50 = percentile fifty; p75 = percentile seventy-five; p90 = percentile ninety; and p97 = percentile ninety-seven.

## Data Availability

The original contributions presented in this study are included in the article. Further inquiries can be directed to the corresponding author.

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
