# Peer review of "Variation in Body Composition Components Across Different Age Groups and Proposal of Age-Specific Normative Tables: A Cross-Sectional Study"

_nutrients, 2025, doi:10.3390/nu17091435_

Round 1
Reviewer 1 Report
Comments and Suggestions for Authors
Title: Body composition components variation in different age groups and a proposal of age-specific normative tables
# This study addresses an important issue related to variations in body composition and provides age-specific normative tables that are valuable for public health and clinical applications. The inclusion of 8,556 individuals strengthens the reliability and generalisability of the findings. The methodology is well described, making the study reproducible. The reference tables developed have considerable potential for use by health professionals.
However, the text contains grammatical errors, awkward phrasing, and wordiness that need refinement; some sections repeat information unnecessarily, making the text less concise; some numerical values and statistical comparisons need clearer explanations; the manuscript has inconsistencies in citations, figure and table references, and in-text formatting; and excessive use of brackets makes the text difficult to follow in some sections.
# Suggestions by lines:
-Lines 15-18. "This study aimed to analyze body composition components in different age groups and to develop age-specific normative tables for Brazilians residing in southern Brazil."
Please, note that "Brazilians residing in southern Brazil" is redundant.
Better: "This study aimed to analyze body composition variations across different age groups and develop age-specific normative tables for individuals in southern Brazil."
-Lines 22-25. "Fat mass and body fat percentage (BFP) were higher in older individuals of both sexes. Among overweight and obese individuals, elevated BFP was particularly noted in obese females aged 40-49 and in males from normal to overweight categories."
The sentence structure is unclear and overly complex.
Better: "Fat mass and body fat percentage (BFP) increased with age in both sexes. Among overweight and obese individuals, BFP was particularly high in obese females aged 40-49 and in
-Lines 35-40. "According to the recent reports from UN, it is expected that, by 2050, the world’s population will reach approximately the 10 billion threshold, representing a threat to the finite planetary resources and intensifying the pressure on natural ecosystems that are already on edge."
Note that "The 10 billion threshold" is awkward and "on edge" is too informal.
Better: "According to recent UN reports, the global population is projected to reach approximately 10 billion by 2050, placing significant strain on finite planetary resources and exacerbating pressure on fragile ecosystems."
-Lines 47-50. "Consequently the seaweed worldwide production witnessed an intensification of more than 3-fold in the period between 2000 and 2020..."
Note that "Witnessed an intensification" is incorrect, awkward phrasing.
Better: "As a result, global seaweed production has increased more than threefold between 2000 and 2020."
-Lines 78-80. "This observational, descriptive, cross-sectional, comparative, and quantitative study."
Note that this phrase is a fragment; lacks a verb.
Better: "This study employed an observational, descriptive, cross-sectional, comparative, and quantitative design."
-Lines 97-100. "Eligible participants were of both sexes, aged 18 to 49 years. A detailed anamnesis was conducted to assess participants' general health, including only those with stable health and without uncontrolled NCDs, such as T2DM or hypertension."
Note that "Including only those..." is unclear.
Better: "Eligible participants were adults aged 18 to 49 years. Only individuals with stable health and without uncontrolled non-communicable diseases (NCDs), such as type 2 diabetes mellitus (T2DM) or hypertension, were included following a detailed health assessment."
-Lines 167-170. "Table 1 presents the general characteristics of the males (n = 3,928 participants), distributed by age group and across age ranges, i.e., 18-29 years, 30-39 years, and 40-49 years."
Note that "Distributed by age group and across age ranges" is redundant.
Better: "Table 1 presents the general characteristics of male participants (n = 3,928) across the age groups: 18-29, 30-39, and 40-49 years."
-Lines 185-189. "It was observed that higher FM in overweight and obese females according to age (Fig. 1A, p<0.01) and in normal and overweight males (Fig. 1B, p<0.001), but not in obese males (Fig. 1B, p<0.01)."
Note that the structure of this sentence is confuse.
Better: "Higher fat mass (FM) was observed with increasing age in overweight and obese females (Fig. 1A, p<0.01) and in normal and overweight males (Fig. 1B, p<0.001), but not in obese males (Fig. 1B, p<0.01)."
-Lines 260-264. "There were progressive higher values for FM and BFP with advancing age in both sexes, which were more pronounced in overweight and obese individuals."
Note that "Progressive higher values" is awkward.
Better: "Fat mass (FM) and body fat percentage (BFP) increased progressively with age in both sexes, with more pronounced effects in overweight and obese individuals."
-Lines 307-310. "This decline in FFM also parallels a reduction in LM and SMM with advancing age, with this diminishment being more pronounced in the older groups."
Note that "With this diminishment being more pronounced" is awkward.
Better: "The decline in FFM is accompanied by reductions in lean mass (LM) and skeletal muscle mass (SMM), particularly in older individuals."
# Suggestions:
-The manuscript needs professional editing to correct grammar, structure and consistency.
- Reduce redundant wording and simplify overly complex sentences.
- Ensure that statistical results are clearly explained.
- Ensure that all citations follow the journal's style guide.
Comments on the Quality of English Language
Please, see report
Author Response
We would like to thank you for the opportunity to submit our revised and corrected manuscript (Nutrients-3526582). We hope that we have met the demands of the reviewers. Should any further changes be necessary, we will be pleased to consider them. Below, we list our responses to the referees’ critical reviews.
Comments and answers:
Reviewer #1
Title: Body composition components variation in different age groups and a proposal of age-specific normative tables
# This study addresses an important issue related to variations in body composition and provides age-specific normative tables that are valuable for public health and clinical applications. The inclusion of 8,556 individuals strengthens the reliability and generalisability of the findings. The methodology is well described, making the study reproducible. The reference tables developed have considerable potential for use by health professionals.
However, the text contains grammatical errors, awkward phrasing, and wordiness that need refinement; some sections repeat information unnecessarily, making the text less concise; some numerical values and statistical comparisons need clearer explanations; the manuscript has inconsistencies in citations, figure and table references, and in-text formatting; and excessive use of brackets makes the text difficult to follow in some sections.
1) -Lines 15-18. "This study aimed to analyze body composition components in different age groups and to develop age-specific normative tables for Brazilians residing in southern Brazil." Please, note that "Brazilians residing in southern Brazil" is redundant.
Better: "This study aimed to analyze body composition variations across different age groups and develop age-specific normative tables for individuals in southern Brazil."
Answers:
We thank the reviewer for this comment. We modified the text according to the reviewer’s suggestion. The changes are highlighted in yellow.
2) -Lines 22-25. "Fat mass and body fat percentage (BFP) were higher in older individuals of both sexes. Among overweight and obese individuals, elevated BFP was particularly noted in obese females aged 40-49 and in males from normal to overweight categories."
The sentence structure is unclear and overly complex.
Better: "Fat mass and body fat percentage (BFP) increased with age in both sexes. Among overweight and obese individuals, BFP was particularly high in obese females aged 40-49 and in
Answers:
We thank the reviewer for the suggestion, and we have corrected the text for better clarity. The changes are highlighted in yellow.
3) -Lines 35-40. "According to the recent reports from UN, it is expected that, by 2050, the world’s population will reach approximately the 10 billion threshold, representing a threat to the finite planetary resources and intensifying the pressure on natural ecosystems that are already on edge."
Note that "The 10 billion threshold" is awkward and "on edge" is too informal.
Better: "According to recent UN reports, the global population is projected to reach approximately 10 billion by 2050, placing significant strain on finite planetary resources and exacerbating pressure on fragile ecosystems."
Answers:
We thank the reviewer for the suggestion, but those lines are probably from another manuscript the reviewer is reviewing.
4) -Lines 47-50. "Consequently the seaweed worldwide production witnessed an intensification of more than 3-fold in the period between 2000 and 2020..."
Note that "Witnessed an intensification" is incorrect, awkward phrasing.
Better: "As a result, global seaweed production has increased more than threefold between 2000 and 2020."
Answers:
We thank the reviewer for the suggestion, but those lines are probably from another manuscript the reviewer is reviewing.
5) -Lines 78-80. "This observational, descriptive, cross-sectional, comparative, and quantitative study."
Note that this phrase is a fragment; lacks a verb.
Better: "This study employed an observational, descriptive, cross-sectional, comparative, and quantitative design."
Answers:
We thank the reviewer for the comment and have corrected the text accordingly. The changes are highlighted in yellow.
6) -Lines 97-100. "Eligible participants were of both sexes, aged 18 to 49 years. A detailed anamnesis was conducted to assess participants' general health, including only those with stable health and without uncontrolled NCDs, such as T2DM or hypertension."
Note that "Including only those..." is unclear.
Better: "Eligible participants were adults aged 18 to 49 years. Only individuals with stable health and without uncontrolled non-communicable diseases (NCDs), such as type 2 diabetes mellitus (T2DM) or hypertension, were included following a detailed health assessment."
Answers:
Thank you for the suggestion. We adjusted the text accordingly. The changes are highlighted in yellow.
7) -Lines 167-170. "Table 1 presents the general characteristics of the males (n = 3,928 participants), distributed by age group and across age ranges, i.e., 18-29 years, 30-39 years, and 40-49 years."
Note that "Distributed by age group and across age ranges" is redundant.
Better: "Table 1 presents the general characteristics of male participants (n = 3,928) across the age groups: 18-29, 30-39, and 40-49 years."
Answers:
We thank the reviewer for the observation and have corrected the text accordingly. The changes are highlighted in yellow.
8) -Lines 185-189. "It was observed that higher FM in overweight and obese females according to age (Fig. 1A, p<0.01) and in normal and overweight males (Fig. 1B, p<0.001), but not in obese males (Fig. 1B, p<0.01)."
Note that the structure of this sentence is confuse.
Better: "Higher fat mass (FM) was observed with increasing age in overweight and obese females (Fig. 1A, p<0.01) and in normal and overweight males (Fig. 1B, p<0.001), but not in obese males (Fig. 1B, p<0.01)."
Answers:
We thank the reviewer for the observation and have corrected the text accordingly. The changes are highlighted in yellow.
9) -Lines 260-264. "There were progressive higher values for FM and BFP with advancing age in both sexes, which were more pronounced in overweight and obese individuals."
Note that "Progressive higher values" is awkward.
Better: "Fat mass (FM) and body fat percentage (BFP) increased progressively with age in both sexes, with more pronounced effects in overweight and obese individuals."
Answers:
We thank the reviewer for the observation and have corrected the text of the manuscript accordingly. The changes are highlighted in yellow.
10) -Lines 307-310. "This decline in FFM also parallels a reduction in LM and SMM with advancing age, with this diminishment being more pronounced in the older groups."
Note that "With this diminishment being more pronounced" is awkward.
Better: "The decline in FFM is accompanied by reductions in lean mass (LM) and skeletal muscle mass (SMM), particularly in older individuals."
Answers:
We thank the reviewer for the suggestion and have corrected the text accordingly. The changes are highlighted in yellow.
# Suggestions:
-The manuscript needs professional editing to correct grammar, structure and consistency.
- Reduce redundant wording and simplify overly complex sentences.
- Ensure that statistical results are clearly explained.
- Ensure that all citations follow the journal's style guide.
Answers:
We thank the reviewer for the suggestions. We thoroughly analyzed the manuscript to improve the readability and clarity of the text.
We would like to thank the reviewers for their feedback. We believe that all points have been addressed and that we are available for any questions or adjustments to the manuscript. The article has undergone extensive grammatical review. We have also recreated the figures. All figures are in high resolution. We have uploaded the figures separately if the editor/reviewers have suggestions for changes or integrations into the article. Thank you in advance for your cooperation.

Reviewer 2 Report
Comments and Suggestions for Authors
This article has not fully answered some of the questions due to insufficient description.
First, authors showed a lot of graphs in figures, but it is difficult to read them, because the letters are too small. Authors should revise the figures, carefully.
Second, authors suggest “This observational, descriptive, cross-sectional, comparative, and quantitative study [20].” (L79) as “2.1. Study Design”, but this description is insufficient. For example, authors should explain sampling method and recruitment method. Authors should revise the study design section, carefully.
Third, authors suggest “Utilizing a representative sample” (L15), but “The sample was restricted to a specific region of Brazil, potentially limiting the generalizability of the findings to other populations.”’ (L327), as mentioned by authors. Moreover, authors could not justify representativeness of samples, because they do not explain sampling methods. Furthermore, authors do not show the flow chart of the number of recruited persons, participants, excluded persons, and persons included in these analyses. Authors should revie the method section, carefully.
Fourth, authors suggest “the sample size calculation for each age group and sex was performed, employing a 99% confidence level with a 5% margin of error.” (L91), but it is difficult for readers to understand how authors calculated it. For example, authors may aim to detect differences among gender and age groups, but they do not show the assumptions of differences. Moreover, the main result this study may be figure 1-5, but they do not use BMI category for sample size calculation. Authors should explain how they calculated sample size in the method section.
Fifth, authors used “T2DM, type 1 diabetes mellitus, hypertension, renal failure, severe liver disease” (L110), but they do not explain their definitions. It is difficult for readers to understand what authors did, without the definitions. Authors should add their definitions in method section.
Sixth, authors explain BMI category as “nutritional status” (L281), but BMI category is not always base on“nutritional status”, as mentioned by authors. Authors should use appropriate terms.
Finally, authors suggest “Based on these results, lifestyle interventions that combine nutrition and physical exercise may be recommended to mitigate these effects of aging.” (L29), but they could not justify these descriptions in this manuscript including the result section. If authors suggest this statement, they should show intervention study in this manuscript. Authors should revise the manuscript, carefully.
Author Response
Reviewer #2
Comments and Suggestions for Authors
This article has not fully answered some of the questions due to insufficient description.
1) First, authors showed a lot of graphs in figures, but it is difficult to read them, because the letters are too small. Authors should revise the figures, carefully.
Answers:
We thank the reviewer for the suggestion and revised all the figures included in the manuscript, improving readability and clarity.
2) Second, authors suggest “This observational, descriptive, cross-sectional, comparative, and quantitative study [20].” (L79) as “2.1. Study Design”, but this description is insufficient. For example, authors should explain sampling method and recruitment method. Authors should revise the study design section, carefully.
Answers:
We thank the reviewer for this comment. With respect to the sampling method, we calculate the minimum sample size via an online sample size calculator (http://www.raosoft.com/samplesize.html) with a 99% confidence level and a 5% margin of error. The minimum sample size is described by sex and age group in section 2.2, Participants, of the manuscript. We will update the description using the tool used for the sampling to clarify this section. Changes are highlighted in yellow.
3) Third, authors suggest “Utilizing a representative sample” (L15), but “The sample was restricted to a specific region of Brazil, potentially limiting the generalizability of the findings to other populations.”’ (L327), as mentioned by authors. Moreover, authors could not justify representativeness of samples, because they do not explain sampling methods. Furthermore, authors do not show the flow chart of the number of recruited persons, participants, excluded persons, and persons included in these analyses. Authors should revie the method section, carefully.
Answers:
We thank the reviewer for this comment. We agree with the reviewer regarding the generalizability of the study. Our sample is from southern Brazil, and other studies in different regions could confirm the validity of our proposed normative table. Nevertheless, because of the n of individuals analyzed in our study, those findings can be confirmed by further studies. With respect to the flowchart of the study, the sampling occurred from 2017--2024. Our study is cross-sectional, and we excluded individuals with the criteria cited in section 2.2 Participants. Because we analyzed the data retrospectively, we focused on the minimum sample size for the validity of the study and did not accompany the participants longitudinally. Considering this, we do not see the need to add a flowchart describing the study.
4) Fourth, authors suggest “the sample size calculation for each age group and sex was performed, employing a 99% confidence level with a 5% margin of error.” (L91), but it is difficult for readers to understand how authors calculated it. For example, authors may aim to detect differences among gender and age groups, but they do not show the assumptions of differences. Moreover, the main result this study may be figure 1-5, but they do not use BMI category for sample size calculation. Authors should explain how they calculated sample size in the method section.
Answers:
The sample size calculation description was added to section 2.2 Participants. The objective of this study was to evaluate whether aging alters body composition parameters and whether those changes are influenced by BMI. The main findings of the study are shown in Figure 6, Table 3 and Table 4. Figure 6 summarizes the main results of the study, and Tables 3 and 4 present the cutoff points for body composition. As we previously noted, we have clarified this section with a description of the sample size.
5) Fifth, authors used “T2DM, type 1 diabetes mellitus, hypertension, renal failure, severe liver disease” (L110), but they do not explain their definitions. It is difficult for readers to understand what authors did, without the definitions. Authors should add their definitions in method section.
Answers:
We excluded those individuals from the study on the basis of a detailed health assessment. The criteria were based on participants' self-reported diagnoses of chronic diseases. We included the criteria of diagnosis utilized for better clarity. Changes are highlighted in yellow.
6) Sixth, authors explain BMI category as “nutritional status” (L281), but BMI category is not always base on“nutritional status”, as mentioned by authors. Authors should use appropriate terms.
Answers:
We thank the reviewer for this comment. We altered the text from “nutritional status” to “BMI” for better clarity. Changes are highlighted in yellow.
7) Finally, authors suggest “Based on these results, lifestyle interventions that combine nutrition and physical exercise may be recommended to mitigate these effects of aging.” (L29), but they could not justify these descriptions in this manuscript including the result section. If authors suggest this statement, they should show intervention study in this manuscript. Authors should revise the manuscript, carefully.
Answers:
We thank the reviewer for the suggestion. We added this point to address the requested question: “Finally, in view of changes in the aging process, regular exercise, particularly resistance training, can normalize some aspects of age-related mitochondrial dysfunction and improve muscle function. Simultaneously, good nutrition, especially adequate protein intake, is essential for limiting and treating age-related decreases in muscle mass, strength, and functional ability. The combination of protein nutrition and exercise is considered optimal for maintaining muscle function, preventing frailty, and sustaining independence during aging [30].
- Deutz, N. E. P.; Bauer, J. M.; Barazzoni, R.; Biolo, G.; Boirie, Y.; Bosy-Westphal, A.; Cederholm, T.; Cruz-Jentoft, A.; Krznariç, Z.; Nair, K. S. Protein intake and exercise for optimal muscle function with aging: recommendations from the ESPEN expert group. Clinical Nutrition 2014, 929-936. Doi: http://dx.doi.org/10.1016/j.clnu.2014.04.007
We would like to thank the reviewers for their feedback. We believe that all points have been addressed and that we are available for any questions or adjustments to the manuscript. The article has undergone extensive grammatical review. We have also recreated the figures. All figures are in high resolution. We have uploaded the figures separately if the editor/reviewers have suggestions for changes or integrations into the article. Thank you in advance for your cooperation.

Round 2
Reviewer 1 Report
Comments and Suggestions for Authors
The manuscript has been significantly improved, so it could be published in its current form.
Author Response
Thank you for your feedback
Reviewer 2 Report
Comments and Suggestions for Authors
Authors revised the manuscript, but this article has not fully answered some of the questions due to insufficient description.
First, as mentioned in the previous review, authors suggest “This study employed an observational, descriptive, cross-sectional, comparative, and quantitative design [20].” (L80) as “2.1. Study Design”, but this description is insufficient. For example, authors should explain sampling method and recruitment method (e.g., cluster sampling). Authors suggest “We thank the reviewer for this comment. With respect to the sampling method, we calculate the minimum sample size via an online sample size calculator (http://www.raosoft.com/samplesize.html) with a 99% confidence level and a 5% margin of error. The minimum sample size is described by sex and age group in section 2.2, Participants, of the manuscript. We will update the description using the tool used for the sampling to clarify this section.”, and they have not responded to any of the comments. Authors should revise the study design section, carefully.
Second, as mentioned in the previous review, authors suggest “Utilizing a representative sample” (L15), but “The sample was restricted to a specific region of Brazil, potentially limiting the generalizability of the findings to other populations.”’ (L337), as mentioned by authors. Authors suggest “We agree with the reviewer regarding the generalizability of the study. Our sample is from southern Brazil, and other studies in different regions could confirm the validity of our proposed normative table. Nevertheless, because of the n of individuals analyzed in our study, those findings can be confirmed by further studies.”, but they have not responded to any of the comments. Moreover, authors could not justify representativeness of samples, because they do not explain sampling methods. Furthermore, authors do not show the flow chart of the number of recruited persons, participants, excluded persons, and persons included in these analyses. Authors suggest “With respect to the flowchart of the study, the sampling occurred from 2017--2024. Our study is cross-sectional, and we excluded individuals with the criteria cited in section 2.2 Participants. Because we analyzed the data retrospectively, we focused on the minimum sample size for the validity of the study and did not accompany the participants longitudinally. Considering this, we do not see the need to add a flowchart describing the study.”, but they do no explain the number of recruited persons, participants, excluded persons, and persons included in these analyses. Authors should revie the method section using flowchart, carefully.
Third, as mentioned in the previous review, authors suggest “the sample size calculation for each age group and sex was performed, employing a 99% confidence level with a 5% margin of error.” (L91), but it is difficult for readers to understand how authors calculated it. For example, authors may aim to detect differences among gender and age groups, but they do not show the assumptions of differences. Authors suggest “The sample size calculation description was added to section 2.2 Participants. The objective of this study was to evaluate whether aging alters body composition parameters and whether those changes are influenced by BMI. The main findings of the study are shown in Figure 6, Table 3 and Table 4. Figure 6 summarizes the main results of the study, and Tables 3 and 4 present the cutoff points for body composition. As we previously noted, we have clarified this section with a description of the sample size.”, but they have not responded to any of the comments.. Moreover, the main result this study may be figure 1-5, but they do not use BMI category for sample size calculation. Authors should explain how they calculated sample size in the method section.
Fourth, authors suggest “T2DM (fasting glycemia ≥ 126 mg/dl), type 1 diabetes mellitus (fasting glycemia ≥ 126 mg/dL),” (L111) to explain their definitions, but it is difficult to identify differences between T2DM and type 1 diabetes mellitus from their definition. Authors should explain their definitions in method section, clearly.
Finally, as mentioned in the previous review, authors suggest “On the basis of these results, lifestyle interventions that combine nutrition and physical exercise may be recommended to mitigate these effects of aging.” (L29), but they could not justify these descriptions in this manuscript based on the result section. Authors suggest “We added this point to address the requested question: “Finally, in view of changes in the aging process, regular exercise, particularly resistance training, can normalize some aspects of age-related mitochondrial dysfunction and improve muscle function. Simultaneously, good nutrition, especially adequate protein intake, is essential for limiting and treating age-related decreases in muscle mass, strength, and functional ability. The combination of protein nutrition and exercise is considered optimal for maintaining muscle function, preventing frailty, and sustaining independence during aging [30].”, but if authors suggest this statement, they should show intervention study in this manuscript. Authors should revise the manuscript, carefully.
Author Response
We would like to thank you for the opportunity of sending our revised and corrected manuscript (nutrients-3526582). We hope that we have met the demands of the reviewers. Should any further changes be necessary, we will be pleased to consider them. Below we listed our responses to the Referees' critical reviews.
Comments and answers:
Reviewer #2
Comments and Suggestions for Authors
Authors revised the manuscript, but this article has not fully answered some of the questions due to insufficient description.
1) First, as mentioned in the previous review, authors suggest "This study employed an observational, descriptive, cross-sectional, comparative, and quantitative design [20]." (L80) as "2.1. Study Design", but this description is insufficient. For example, authors should explain sampling method and recruitment method (e.g., cluster sampling). Authors suggest "We thank the reviewer for this comment. With respect to the sampling method, we calculate the minimum sample size via an online sample size calculator (http://www.raosoft.com/samplesize.html) with a 99% confidence level and a 5% margin of error. The minimum sample size is described by sex and age group in section 2.2, Participants, of the manuscript.
We will update the description using the tool used for the sampling to clarify this section.", and they have not responded to any of the comments. Authors should revise the study design section, carefully.
Answers:
We thank the reviewer for the suggestion. We adjusted to: This study employed an observational, descriptive, cross-sectional, comparative, and quantitative design [20] following the Strengthening the Reporting of Observational Studies in Epidemiology (STROBE) Statement [21]. It was utilized a cluster sampling method for the recruitment of participants via Municipal Hospital, "Acclimation" Basic Health Unit (located inside the University Center), and Interdisciplinary Laboratory for Intervention in Health Promotion (the last from social media platforms), all the means by inclusion and exclusion criteria, recruiting a similar number of people in each recruitment center.
2) Second, as mentioned in the previous review, authors suggest "Utilizing a representative sample" (L15), but "The sample was restricted to a specific region of Brazil, potentially limiting the generalizability of the findings to other populations."’ (L337), as mentioned by authors. Authors suggest “We agree with the reviewer regarding the generalizability of the study. Our sample is from southern Brazil, and other studies in different regions could confirm the validity of our proposed normative table. Nevertheless, because of the n of individuals analyzed in our study, those findings can be confirmed by further studies.”, but they have not responded to any of the comments. Moreover, authors could not justify representativeness of samples, because they do not explain sampling methods. Furthermore, authors do not show the flow chart of the number of recruited persons, participants, excluded persons, and persons included in these analyses. Authors suggest “With respect to the flowchart of the study, the sampling occurred from 2017--2024. Our study is cross-sectional, and we excluded individuals with the criteria cited in section 2.2 Participants. Because we analyzed the data retrospectively, we focused on the minimum sample size for the validity of the study and did not accompany the participants longitudinally. Considering this, we do not see the need to add a flowchart describing the study.”, but they do no explain the number of recruited persons, participants, excluded persons, and persons included in these analyses. Authors should revie the method section using flowchart, carefully.
Answers:
We thank the reviewer for the comment. We included the STROBE flowchart describing the study's design in section 2.2 Participants.
3) Third, as mentioned in the previous review, authors suggest “the sample size calculation for each age group and sex was performed, employing a 99% confidence level with a 5% margin of error.” (L91), but it is difficult for readers to understand how authors calculated it. For example, authors may aim to detect differences among gender and age groups, but they do not show the assumptions of differences. Authors suggest “The sample size calculation description was added to section 2.2 Participants. The objective of this study was to evaluate whether aging alters body composition parameters and whether those changes are influenced by BMI. The main findings of the study are shown in Figure 6, Table 3 and Table 4. Figure 6 summarizes the main results of the study, and Tables 3 and 4 present the cutoff points for body composition. As we previously noted, we have clarified this section with a description of the sample size.”, but they have not responded to any of the comments. Moreover, the main result this study may be figure 1-5, but they do not use BMI category for sample size calculation. Authors should explain how they calculated sample size in the method section.
Answers:
We thank the reviewer for this comment. We aimed to detect body composition differences according to gender, age, and BMI. As described in the methods section, the minimum sample size was around 650 individuals per group when we inferred the minimum sample size with 99% confidence intervals and 5% margin of error. Here’s the table describing the count discriminated by age, BMI, and Gender.
|
Age |
BMI |
Gender |
Count |
|
18 to 29 |
Normal |
Female |
1303 |
|
30 to 39 |
Normal |
Female |
575 |
|
40 to 49 |
Normal |
Female |
363 |
|
18 to 29 |
Overweight |
Female |
629 |
|
30 to 39 |
Overweight |
Female |
400 |
|
40 to 49 |
Overweight |
Female |
400 |
|
18 to 29 |
Obese |
Female |
242 |
|
30 to 39 |
Obese |
Female |
352 |
|
40 to 49 |
Obese |
Female |
449 |
|
18 to 29 |
Normal |
Male |
994 |
|
30 to 39 |
Normal |
Male |
314 |
|
40 to 49 |
Normal |
Male |
128 |
|
18 to 29 |
Overweight |
Male |
850 |
|
30 to 39 |
Overweight |
Male |
536 |
|
40 to 49 |
Overweight |
Male |
409 |
|
18 to 29 |
Obese |
Male |
221 |
|
30 to 39 |
Obese |
Male |
247 |
|
40 to 49 |
Obese |
Male |
225 |
If we make a power simulation to achieve 80% power to detect a small effect size (f=0.10) in an ANOVA with 18 groups at alpha=0.05, we would need at least 1,985 participants, so our study is well-powered for its objectives. Below is the simulation of statistical power.
We updated the methods and included a 2.2 section on Statistical power analysis to clarify the sample size calculation. Changes are highlighted in yellow.
4) Fourth, authors suggest “T2DM (fasting glycemia ≥ 126 mg/dl), type 1 diabetes mellitus (fasting glycemia ≥ 126 mg/dL),” (L111) to explain their definitions, but it is difficult to identify differences between T2DM and type 1 diabetes mellitus from their definition. Authors should explain their definitions in method section, clearly.
Answers:
The exclusion criteria were pregnant and menstruating females (during assessment), individuals with pacemakers or other implantable electronic devices, patients with paraplegia or quadriplegia, cancer, or bariatric surgery, and patients with noncontrolled chronic diseases (i.e., T2DM (fasting glycemia ≥ 126 mg/dL), type 1 diabetes mellitus (fasting glycemia ≥ 126 mg/dL), hypertension (blood pressure ≥ 130/80 mmHg), renal failure (glomerular filtration rate (GFR) < 15 mL/min), or severe liver disease (diagnosis of cirrhosis), where all exams were requested a prior for the patients). Thus, the final sample did not include individuals reporting such criteria. The inclusion and exclusion criteria are summarized in Figure 1.
5) Finally, as mentioned in the previous review, authors suggest “On the basis of these results, lifestyle interventions that combine nutrition and physical exercise may be recommended to mitigate these effects of aging.” (L29), but they could not justify these descriptions in this manuscript based on the result section. Authors suggest “We added this point to address the requested question: “Finally, in view of changes in the aging process, regular exercise, particularly resistance training, can normalize some aspects of age-related mitochondrial dysfunction and improve muscle function. Simultaneously, good nutrition, especially adequate protein intake, is essential for limiting and treating age-related decreases in muscle mass, strength, and functional ability. The combination of protein nutrition and exercise is considered optimal for maintaining muscle function, preventing frailty, and sustaining independence during aging [30].”, but if authors suggest this statement, they should show intervention study in this manuscript. Authors should revise the manuscript, carefully.
Answers:
We thank the reviewer for the suggestion. We included citations of intervention studies in the main text. Changes are highlighted in yellow.
PS: PLEASE, SEE OUR ATTACHED DOCUMENT.

Round 3
Reviewer 2 Report
Comments and Suggestions for Authors
Authors revised the manuscript, but this article has not fully answered some of the questions due to insufficient description.
First, as mentioned in the previous reviews, the description regarding sampling method is insufficient. Authors added the descriptions “This study employed an observational, descriptive, cross-sectional, comparative, and quantitative design [22] following the Strengthening the Reporting of Observational Studies in Epidemiology (STROBE) Statement [23]. It was utilized a cluster sampling method for the recruitment of participants via the Municipal Hospital, "Acclimation" Basic Health Unit (located inside the University Center), and Interdisciplinary Laboratory for Intervention in Health Promotion (the last from social media platforms), all the means by inclusion and exclusion criteria, recruiting a similar number of people in each recruitment center.” (L84), but if authors used “a cluster sampling method”, they should explain the number of clusters, response rate, and such. It is also important to use multilevel design to reduce the bias from a cluster sampling method and imbalance of response rate among clusters. Authors should revise the study design section as well as statistical methods, carefully.
Second, as mentioned in the previous review, authors suggest “Utilizing a representative sample” (L16) and “, this research has several strengths, as follows: a large sample size, which is representative of Paraná State;” (L361), but “The sample was restricted to a specific region of Brazil”’ (L349), as mentioned by authors. Authors also suggest “the recruitment of participants via the Municipal Hospital,”, which means that this sampling may not be population-based. Authors should revise the manuscript, carefully.
Third, authors added the descriptions “80% power was used to detect a minimum effect size (f=0.10) in an ANOVA, including 18 groups at an alpha level of 0.05.” (L93), but they do not explain “18 groups”. Authors should add explanation regarding “18 groups”.
Fourth, authors suggest “T2DM (fasting glycemia ≥ 126 mg/dl), type 1 diabetes mellitus (fasting glycemia ≥ 126 mg/dL),” (L111) to explain their definitions, but it is difficult to identify differences between T2DM and type 1 diabetes mellitus from their definition. Authors suggest “The exclusion criteria were pregnant and menstruating females (during assessment), individuals with pacemakers or other implantable electronic devices, patients with paraplegia or quadriplegia, cancer, or bariatric surgery, and patients with noncontrolled chronic diseases (i.e., T2DM (fasting glycemia ≥ 126 mg/dL), type 1 diabetes mellitus (fasting glycemia ≥ 126 mg/dL), hypertension (blood pressure ≥ 130/80 mmHg), renal failure (glomerular filtration rate (GFR) < 15 mL/min), or severe liver disease (diagnosis of cirrhosis), where all exams were requested a prior for the patients). Thus, the final sample did not include individuals reporting such criteria. The inclusion and exclusion criteria are summarized in Figure 1.”, but and they have not responded to any of the comments. Authors should explain their definitions in method section, clearly.
Fifth, authors suggest “A total of 11,654 individuals were recruited. Following the initial screening, 1,596 individuals were excluded, leaving 10,058 participants. After a second screening, 1,452 participants were excluded from the study.” (L98), but they do not explain about “the initial screening” and “a second screening”. Authors should add the descriptions regarding explanation about “the initial screening” and “a second screening” in method section.
Finally, as mentioned in the previous review, authors suggest “. Based on these results, lifestyle interventions that combine nutrition and physical exercise may be recommended to mitigate these effects of aging.” (L31), but they could not justify these descriptions in this manuscript based on the result section. Authors suggest “We included citations of intervention studies in the main text.”, but if authors suggest this statement, they should show intervention study in this manuscript. Authors should revise the manuscript, carefully.
Author Response
Dear editor and reviewer, we have answered all the questions you asked. Below are our responses. Thank you.
Authors revised the manuscript, but this article has not fully answered some of the questions due to insufficient description.
RE: We adjusted all the points requested. Thank you for your note.
First, as mentioned in the previous reviews, the description regarding sampling method is insufficient. Authors added the descriptions “This study employed an observational, descriptive, cross-sectional, comparative, and quantitative design [22] following the Strengthening the Reporting of Observational Studies in Epidemiology (STROBE) Statement [23].
RE: No changes at this point.
It was utilized a cluster sampling method for the recruitment of participants via the Municipal Hospital, "Acclimation" Basic Health Unit (located inside the University Center), and Interdisciplinary Laboratory for Intervention in Health Promotion (the last from social media platforms), all the means by inclusion and exclusion criteria, recruiting a similar number of people in each recruitment center.” (L84), but if authors used “a cluster sampling method”, they should explain the number of clusters, response rate, and such.
RE: It was employed a cluster sampling method utilizing three clusters: the Municipal Hospital, the 'Acclimation' Basic Health Unit, and the Interdisciplinary Laboratory for Intervention in Health Promotion. The response rate was similar across the groups, as we recruited the 11,554 participants homogeneously, with one participant each from the Municipal Hospital, the Basic Health Unit, and the Interdisciplinary Laboratory, continuing this pattern until reaching the 8,556 eligible participants. Consequently, 3,108 participants were deemed ineligible based on exclusion criteria (conforming Figure 1).
It is also important to use multilevel design to reduce the bias from a cluster sampling method and imbalance of response rate among clusters. Authors should revise the study design section as well as statistical methods, carefully.
RE: "We thank the reviewer for the suggestion regarding the potential impact of our cluster sampling design and the recommendation to consider a multilevel approach. We acknowledge that cluster sampling can introduce dependencies that may require specific analytical adjustments.
We performed an empirical assessment to quantify the degree of clustering present in our data specifically related to our outcome variables. We calculated the Intraclass Correlation Coefficient (ICC) for our primary body composition outcomes (Body Fat Mass, Lean Mass, Fat-Free Mass, Skeletal Muscle Mass, and Percent Body Fat), using the recruitment site (the Municipal Hospital, the 'Acclimation' Basic Health Unit, and the Interdisciplinary Laboratory for Intervention in Health Promotion) as the clustering factor.
The results unequivocally showed negligible clustering effects for all measured body composition parameters. Specifically, the adjusted ICC was 0.000 for each of these variables:
Body Fat Mass (ICC = 0.000)
Lean Mass (ICC = 0.000)
Fat-Free Mass (ICC = 0.000)
Skeletal Muscle Mass (ICC = 0.000)
Percent Body Fat (ICC = 0.000)
An ICC of 0.000 indicates that virtually none of the observed variance in these body composition measures is attributable to differences between the recruitment clusters. This finding strongly suggests that the potential violation of the independence assumption, which multilevel modeling aims to correct, is not a substantive issue for these specific outcomes in our dataset. Given this clear empirical evidence, employing a multilevel model (or other cluster-robust adjustments) would introduce unnecessary complexity without providing a statistically meaningful correction or altering the conclusions regarding our primary research questions.
We will revise the Statistical Methods section of the manuscript to explicitly report this ICC analysis, present the ICC=0.000 results, and provide this empirical justification for not employing a multilevel modeling framework for these specific analyses. Changes are highlighted in yellow.
Second, as mentioned in the previous review, authors suggest “Utilizing a representative sample” (L16) and “, this research has several strengths, as follows: a large sample size, which is representative of Paraná State;” (L361), but “The sample was restricted to a specific region of Brazil”’ (L349), as mentioned by authors. Authors also suggest “the recruitment of participants via the Municipal Hospital,”, which means that this sampling may not be population-based. Authors should revise the manuscript, carefully.
RE: It was changed. We changed “representative sample” to “significative sample”; and excluded the words: “which is representative of Paraná State”. Thank you for your appointment.
Third, authors added the descriptions “80% power was used to detect a minimum effect size (f=0.10) in an ANOVA, including 18 groups at an alpha level of 0.05.” (L93), but they do not explain “18 groups”. Authors should add explanation regarding “18 groups”.
RE: We adjusted this point to: “A power analysis with the following parameters was performed to ensure adequate statistical power: 80% power was used to detect a minimum effect size (f=0.10) in an analysis of variance (ANOVA), including 18 groups at an alpha level of 0.05. The '18 groups' refer to the different combinations of age and BMI categories across sexes. Specifically, the groups were as follows: individuals aged 19-29 with normal BMI, over-weight, and obesity; individuals aged 30-39 with the same BMI categories; and individuals aged 40-49 also with the same conditions. Each of these age and BMI categories was further divided by sex (male and female), resulting in the 18 distinct groups analyzed. The resulting minimum required sample size was calculated to be 1,985 individuals. The analysis was done in R (version 4.4.2) and RStudio (version 2024.09.1+394).”
Fourth, authors suggest “T2DM (fasting glycemia ≥ 126 mg/dl), type 1 diabetes mellitus (fasting glycemia ≥ 126 mg/dL),” (L111) to explain their definitions, but it is difficult to identify differences between T2DM and type 1 diabetes mellitus from their definition. Authors suggest “The exclusion criteria were pregnant and menstruating females (during assessment), individuals with pacemakers or other implantable electronic devices, patients with paraplegia or quadriplegia, cancer, or bariatric surgery, and patients with noncontrolled chronic diseases (i.e., T2DM (fasting glycemia ≥ 126 mg/dL), type 1 diabetes mellitus (fasting glycemia ≥ 126 mg/dL), hypertension (blood pressure ≥ 130/80 mmHg), renal failure (glomerular filtration rate (GFR) < 15 mL/min), or severe liver disease (diagnosis of cirrhosis), where all exams were requested a prior for the patients). Thus, the final sample did not include individuals reporting such criteria. The inclusion and exclusion criteria are summarized in Figure 1.”, but and they have not responded to any of the comments. Authors should explain their definitions in method section, clearly.
RE: We added this phrase: “A simple interview was conducted to differentiate between type 1 and type 2 diabetes, focusing on age at diagnosis, treatment history, and the presence of autoimmune markers. This ensured clearer categorization for exclusion.”
Fifth, authors suggest “A total of 11,654 individuals were recruited. Following the initial screening, 1,596 individuals were excluded, leaving 10,058 participants. After a second screening, 1,452 participants were excluded from the study.” (L98), but they do not explain about “the initial screening” and “a second screening”. Authors should add the descriptions regarding explanation about “the initial screening” and “a second screening” in method section.
RE: Thank you for your appointment. We changed to: A total of 11,654 individuals were recruited. During the initial screening, participants were selected based on being within the appropriate age range and expressing interest in the study. At this stage, we did not conduct interviews related to personal histories or pre-existing conditions. Following this initial screening, 1,596 individuals were excluded, leaving 10,058 participants. In the second screening, a more detailed evaluation was conducted, which led to the exclusion of individuals over the age of 50. This decision was made because the analysis lacked sufficient power to include these participants, resulting in 1,452 additional exclusions.
Finally, as mentioned in the previous review, authors suggest “. Based on these results, lifestyle interventions that combine nutrition and physical exercise may be recommended to mitigate these effects of aging.” (L31), but they could not justify these descriptions in this manuscript based on the result section. Authors suggest “We included citations of intervention studies in the main text.”, but if authors suggest this statement, they should show intervention study in this manuscript.
RE: In the discussion section, we suggest that lifestyle interventions combining nutrition and physical exercise may mitigate the effects of aging. This aligns with existing literature and is supported by cited intervention studies within the manuscript. Although the current study focuses on observational findings, these studies provide a basis for our recommendations.
To complement this point, Amaral et al. [29] emphasized maintaining muscle mass to preserve functionality and independence, highlighting the need for preventive strategies such as physical exercise, mainly strength training, along with adequate nutritional support.
We ensure that references to these intervention studies are clearly outlined in the results section to support our statements on lifestyle interventions.
Authors should revise the manuscript, carefully.
RE: We appreciate your feedback and are at your disposal.
